# Uncertainties in deforestation emission baseline methodologies and implications for carbon markets

Hoong Chen Teo [1,2] ✉, Nicole Hui Li Tan[1,2], Qiming Zheng [1,2,3], Annabel Jia Yi Lim[1,2], Rachakonda Sreekar[1,2,4], Xiao Chen[2,5], Yuchuan Zhou [5], Tasya Vadya Sarira[1,2,6], Jose Don T. De Alban [1,2], Hao Tang[2,5], Daniel A. Friess[2,5,7] & Lian Pin Koh [1,2,5,8] ✉

Carbon credits generated through jurisdictional-scale avoided deforestation projects require accurate estimates of deforestation emission baselines, but there are serious challenges to their robustness. We assessed the variability, accuracy, and uncertainty of baselining methods by applying sensitivity and variable importance analysis on a range of typically-used methods and parameters for 2,794 jurisdictions worldwide. The median jurisdiction's deforestation emission baseline varied by 171% (90% range: 87%-440%) of its mean, with a median forecast error of 0.778 times (90% range: 0.548-3.56) the actual deforestation rate. Moreover, variable importance analysis emphasised the strong influence of the deforestation projection approach. For the median jurisdiction, 68.0% of possible methods (90% range: 61.1%-85.6%) exceeded 15% uncertainty. Tropical and polar biomes exhibited larger uncertainties in carbon estimations. The use of sensitivity analyses, multi-model, and multi-source ensemble approaches could reduce variabilities and biases. These findings provide a roadmap for improving baseline estimations to enhance carbon market integrity and trust.

Forest loss accounts for up to one-fifth of anthropogenic carbon emissions to the atmosphere, making it the second largest anthropogenic emissions source after fossil fuel combustion[1,2]. Forest loss also results in the reduction of habitats, biodiversity, and ecosystem services provided by forests such as pollination and climate regulation[3–5], while posing a threat to the livelihoods and cultures of forest-dwelling communities[6,7]. With forest loss projected to continue at a rate of 5.9 Mha y$^{-1}$ without additional action[8], avoiding deforestation is one of the most crucial actions for climate mitigation and sustainable development.

The necessity of mobilising financial support for actions to reduce deforestation is well-recognised, but global efforts at the scale needed have repeatedly stumbled. Initially, policy concerns and technical challenges prevented the inclusion of avoided deforestation projects in the 1997 Kyoto Protocol's Clean Development Mechanism. Efforts to make progress on these challenges culminated in the United Nations Framework Convention for Climate Change's (UNFCCC) Reducing Emissions from Deforestation and forest Degradation (REDD + ) programme, and avoided deforestation projects were officially adopted in 2013. Although the REDD+ framework supports both site-based and

¹Department of Biological Sciences, National University of Singapore, Singapore, Singapore. ²Centre for Nature-based Climate Solutions, National University of Singapore, Singapore, Singapore. ³Department of Land Surveying and Geo-Informatics, Hong Kong Polytechnic University, Hung Hom, Kowloon, Hong Kong SAR. ⁴School of the Environment, University of Queensland, Brisbane, Queensland, Australia. ⁵Department of Geography, National University of Singapore, Singapore, Singapore. ⁶Nicholas School of the Environment, Duke University, Durham, NC, USA. ⁷Department of Earth and Environmental Sciences, Tulane University, New Orleans, LA, USA. ⁸Tropical Marine Science Institute, National University of Singapore, Singapore, Singapore. ✉e-mail: hcteo@u.nus.edu; lianpinkoh@nus.edu.sg

jurisdictional approaches which apply across a national or subnational administrative unit, site-based approaches have been predominant, operating primarily through voluntary carbon markets[9]. In recent years, voluntary carbon markets have expanded rapidly to reach over $2 billion by 2021, with REDD+ projects being the largest by traded volume at $863 million[10]. However, concerns regarding the credibility of avoided deforestation carbon credits have once again risen to the forefront[11], as contrasting evidence on their effectiveness has cast doubt on the methods used to estimate deforestation emission baselines, which define the business-as-usual scenario upon which the emission reductions and subsequent carbon credits are calculated[12,13]. This has posed challenges for carbon market standards, which include the Verified Carbon Standard (VCS), Forest Carbon Partnership Facility (FCPF), and the Architecture for REDD+ Transactions: The REDD+ Environmental Excellence Standard (ART TREES), among others. To maintain credibility, carbon market standards are typically designed to be conservative, through mechanisms such as selecting methodologies with conservative assumptions, and requiring deductions for permanence, leakage, and uncertainty[14,15]. Overestimated baselines may generate credits that lack actual emission reductions, thereby resulting in inefficient resource allocation; conversely, underestimated baselines may result in insufficient financial incentives for forest protection. Thus, uncertainties in baseline projections have important implications for avoided deforestation schemes[16].

Avoided deforestation projects are required to calculate deforestation emission baselines and their corresponding uncertainties following prescribed methods and guidelines (see Methods for details), such as the VCS, FCPF, and ART TREES standards in the voluntary carbon market. Typically, the deforestation emission baseline is a product of two components – a projected deforestation rate for the future period and a forest carbon estimate. Although baselining methods are broadly similar across various standards such as the VCS, FCPF, and ART TREES, even within the same standard a wide range of methods, parameters, and datasets are permitted, leading to concerns about baseline inflation and the overgeneration of credits[17,18]. This also leads to confusion and hesitancy among market participants regarding the reliability of standards and potential future reputational risks of being involved in a project that later becomes controversial[12,19]. Recognising these risks, carbon project developers are working to overhaul standards and address growing concerns about carbon credit integrity[20]. Project developers are also making a stronger push towards jurisdictional and nested REDD+ baselines[21,22], an approach where baselines are determined at jurisdictional-level and applied to jurisdictional projects or allocated to site-based projects nested within the jurisdiction. Although jurisdictional baselines intend to ensure consistency by preventing individual project developers from deliberately making methodological choices to inflate their baselines, they still follow the same broad methodological principles as site-specific baselines. Therefore, jurisdictional baselines are potentially also able to make a wide range of methodological choices, with unknown consequences for the consistency and accuracy of baselines derived.

In order to provide support for enhancing carbon market integrity by improving baselining methods and assist carbon market participants in decision-making, here we assess the variability, accuracy, and uncertainties of baselining methods at the jurisdictional-level on a global scale. Firstly, we assess the range of commonly used methods for establishing deforestation emission baselines, deriving approximately $4 \times 10^9$ unique combinations of methods and parameters which were then applied to calculate baselines for each national or subnational jurisdiction. Secondly, we perform sensitivity analysis to quantify the relative variability of deforestation emission baselines across different methods. Thirdly, we validate the accuracy of different deforestation projection methods against historical data. We also use variable importance analysis to identify the key contributors to the observed relative variability and accuracy. Lastly, we assess

uncertainties inherent to different deforestation projection methods and different models of carbon estimation, and propagate uncertainties in deforestation emission baselines.

## Results

### Relative variability of mean deforestation emission baselines

We estimated the relative variability (as measured by the coefficient of variance, CV = standard deviation/mean) of deforestation emission baselines across all methods for each jurisdiction (Fig. 1). The median variability was 171% (90% range: 87–440%), i.e. using different deforestation emission baselining methods would cause deforestation emission baselines to vary by 171% of the mean deforestation emission baseline for the median jurisdiction. Jurisdictions with high relative variabilities could be found across biomes and latitudes, including in jurisdictions with high forest area and high deforestation rates (Figs. S1–2). There was a significant negative correlation between log-transformed forest area and relative variability, with Pearson's $r = -0.34$ and $p < 0.01$ (Fig. S3a), suggesting that although jurisdictions with larger forest area would generally have less variability in baselines, there were also other factors driving that variability.

Next, we analysed variable importance for how each parameter type influences the deforestation emission baseline (Table 1). Projection approach was the most important variable, followed by the forest dataset used for estimating deforestation. Within each parameter type, we calculated and plotted the relative variabilities of each level (i.e. CV of possible combinations of methods which use only that level) for each jurisdiction (Fig. 2), and performed one-way ANOVAs and Tukey's HSD post-hoc tests to determine which levels had lower relative variabilities and would thus generate more consistent results.

Of the deforestation projection approaches, linear regression models (*global, global_s, regional, and regional_s*; see Methods for explanation) generated the least variable results (Table 1, Fig. 2a); although permitted by VCS standards such as VM0015, these are seldom used compared to the simpler historical average (*hist*) or time function (*linear* and *poly2*) methods (Supplementary Data 1). We found that multi-model ensembles taking the means of 300 linear models which used random combinations of between 3 and 11 driver variables (*global_s* and *regional_s*) had significantly lower median relative variabilities than the corresponding linear models which used all 12 driver variables (*global* and *regional*). Moreover, the multi-model ensembles reduces the interquartile range (IQR) as well as outliers among jurisdictions. However, the regional models had significantly higher median relative variabilities (*regional*: 70.7%, *regional_s*: 65.4%) than their corresponding global models (*global*: 66.0%, *global_s*: 61.0%).

Of the forest datasets used for deforestation projection, Hansen et al.[23] generated the least variable results, regardless of which tree cover threshold was used (Table 1, Fig. 2a). Hansen et al.[23] is also the forest dataset with the highest spatial resolution (30 m) used in this study; however, MODIS[24] (500 m) was observed to generate less variable results than ESA-CCI[25] (300 m). A longer historical reference period, which ranged between 5 and 15 years in this study, also significantly reduced median relative variabilities (Table 1, Fig. 2b); this suggests that longer historical reference period can reduce the effect of stochastic variations in deforestation rates.

Although using different forest datasets used for carbon estimation (forest mask) had statistically significant differences in relative variabilities of the deforestation emission baseline, none of the post-hoc pairwise comparisons were statistically significant. For above-ground biomass carbon (AGBC), data from Soto-Navarro et al.[26]. and Spawn et al.[27], which are both global wall-to-wall products, generate significantly less variable results than the Global Ecosystem Dynamics Investigation (GEDI) product which provides limited sampling tracks[28]. For belowground biomass carbon (BGBC), Soto-Navarro et al.[26] generated the least variable results, and for soil organic carbon (SOC) neither dataset generated less variable results than the other (Table 1).

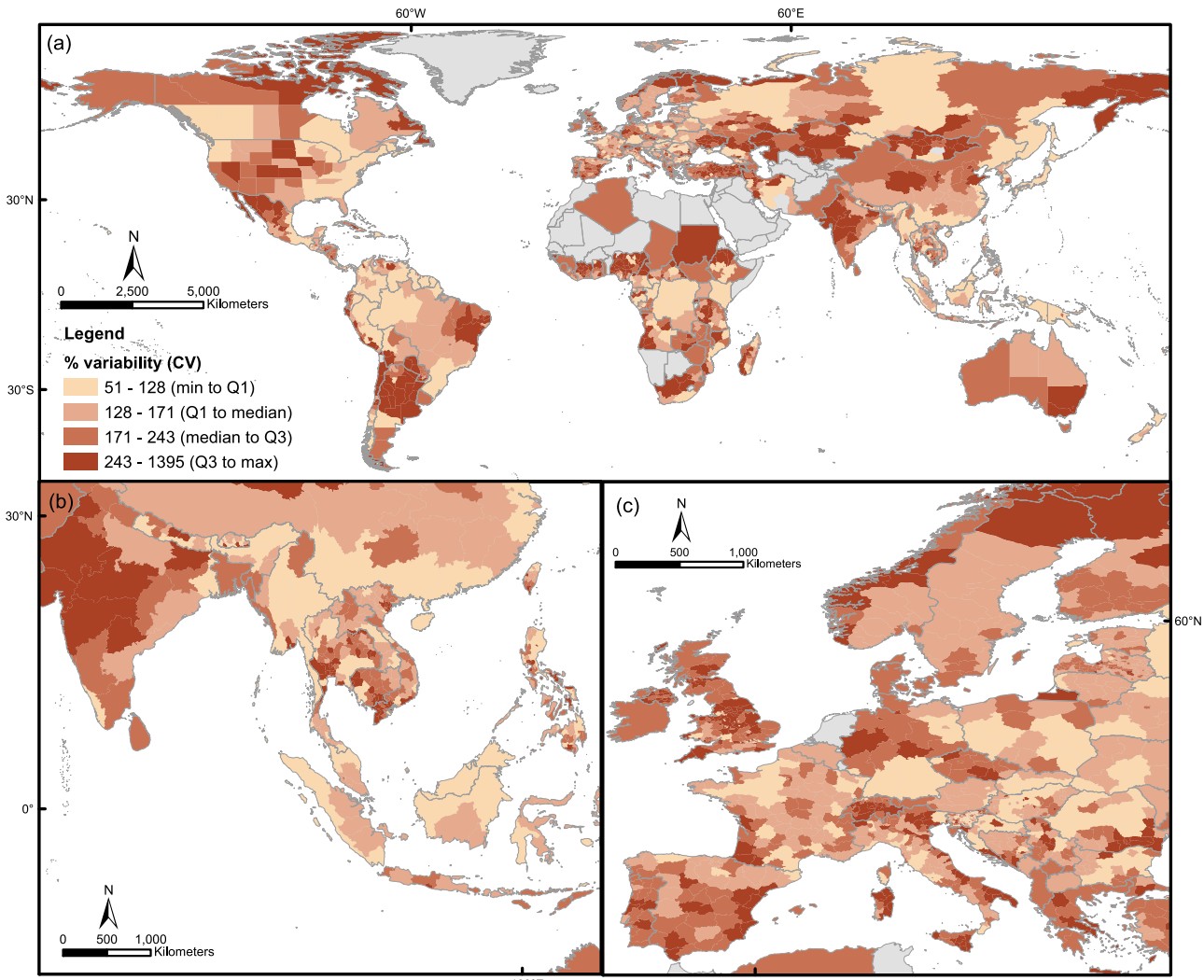

**Fig. 1 | Relative variability of deforestation emission baselines across jurisdictions globally. a** Global map of national or subnational jurisdictions, with colour indicating relative variability of deforestation emissions baselines for each jurisdiction. Deforestation emission baselines are the product of the projected deforestation rate and average forest carbon in $CO_2$e per jurisdiction. **b** Inset of **a** for South and Southeast Asia. **c** Inset of **a** for Europe.

## Historical accuracy of predicted deforestation rates

We assessed the accuracy of predicted deforestation rates by calculating the forecast error (relative difference between actual and predicted deforestation rates); this was achieved by comparing mean deforestation rates during the historical reference period with the ensuing period of the same duration, if the data was available. The median jurisdiction's median forecast error between actual and predicted deforestation (calculated by $\frac{|predicted-actual|}{actual}$) was 0.778 (90% range: 0.548–3.56) across all methods (Fig. 3a), i.e. median predicted deforestation was different from actual deforestation by 0.778 times the actual deforestation. Each projection approach had different spatial variabilities and biases in how accurately they predicted deforestation rates (Fig. S4), without there being clear regional patterns. Forecast errors for the linear models (*global, global_s, regional,* and *regional_s*) were significantly lower than other approaches, but there were no significant differences between the different linear models. However, by taking the median of all approaches, it is possible to overcome the individual spatial variabilities and biases that are inherent to each individual projection approach. Variable importance analysis showed that the projection approach was the most important in affecting accuracy, followed by the forest dataset used, and finally the length of the historical reference period (Table 2). Linear models were

more accurate in predicting deforestation than other approaches (Fig. 4a, S4e-S4f). Hansen et al.[23] was the most historically accurate forest dataset. In general, increasing the length of the historical reference period improves accuracy, resulting in lower medians and IQRs for forecast errors (Fig. 4).

## Uncertainties of deforestation emission baselines, deforestation projections and carbon estimates

We propagated the deforestation projection model uncertainties and carbon pool uncertainties by summation of quadrature (Figs. 3b–f and 4c, d). The median jurisdiction's propagated uncertainty was 29.1% (90% range: 20.8–42.6%) of its deforestation emission baseline. Deforestation projection model uncertainty (median 25.3%; 90% range: 10.1–40.4%) was higher than carbon uncertainty (median 10.7%; 90% range: 6.2–19.5%). There was no clear spatial pattern for overall uncertainty and deforestation projection uncertainty (Fig. 3b-c), but carbon uncertainty appears to be highest in the polar and boreal regions, as well as the tropics (Fig. 3d); in the tropics this was mainly due to AGBC uncertainties (Fig. 3e), while in the Congo basin, polar and boreal regions this was due to both AGBC and BGBC uncertainties (Fig. 3e, f), suggesting systematic issues such as under-sampling of these regions[29]. Voluntary carbon market standards typically require

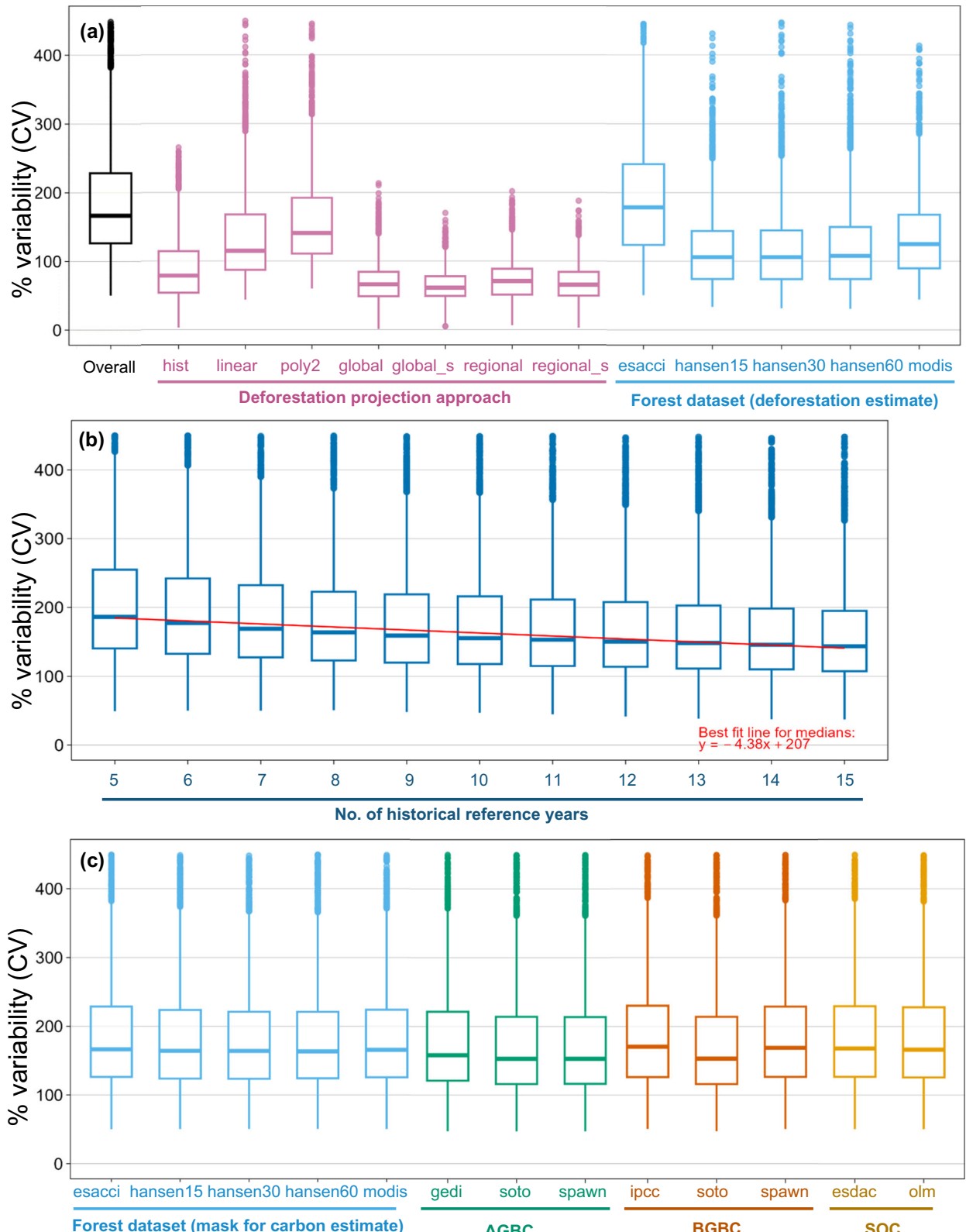

**Fig. 2 | Relative variability of deforestation emission baseline components.** Box plots showing distribution of relative variabilities (% CV) of deforestation emission baselines (each point represents one jurisdiction, for $n = 2794$ jurisdictions) for different component parameters (see explanation of these datasets and methods in Method section): **a** Overall relative variability (across all methods), relative variability for each deforestation projection approach, as well as for each forest dataset used for deforestation projection estimates. Note that *hist* is a simple historical average, *linear* and *poly2* are time functions; *global, global_s, regional,* and *regional_s* are linear regression models. **b** Relative variability for each of the 5–15 possible lengths of historical reference periods used for deforestation projection estimates. **c** Relative variability for each forest dataset used for carbon estimates, as well as each AGBC, BGBC, and SOC dataset.

**Table 1 | Variable importance for how each parameter type influences the deforestation emission baseline, as well as statistical tests (one-way ANOVA and Tukey's HSD post-hoc) comparing relative variabilities between the different levels of each parameter, for $n = 2794$ jurisdictions**

| Parameter | Variable Importance | | Statistical tests | |
|---|---|---|---|---|
| | Mean | SE | One-Way ANOVA | Tukey's HSD post-hoc |
| *Deforestation rate* | | | | |
| Projection approach | 10.2 | 0.18 | $F_{(6,19524)} = 1678$, ***p* < 0.01** | All pairs ***p* < 0.05**, except *global-regional_s* (*p* = 0.1) and *global_s-regional_s* (*p* = 0.07) |
| Forest dataset | 4.41 | 0.14 | $F_{(4,12791)} = 430$, ***p* < 0.01** | All pairs ***p* < 0.05**, except *hansen15-30* (*p* = 0.97) |
| Historical reference years | 2.22 | 0.16 | $F_{(10,30749)} = 82$, ***p* < 0.01** | All pairs ***p* < 0.05**, except *7-8, 8-9, 9-10, 9-11, 10-11, 10-12, 11-12, 11-13, 12-13, 12-14, 13-14, 13-15, 14-15.* |
| *Carbon* | | | | |
| Forest dataset | 0.74 | 0.04 | $F_{(4,13565)} = 2.5$, ***p* = 0.042** | All pairs *p* > 0.05. |
| AGB | 1.31 | 0.04 | $F_{(2,8379)} = 8.4$, ***p* < 0.01** | *gedi-soto* & *gedi-spawn* (***p* < 0.01**); *soto-spawn* (*p* = 0.99) |
| BGB | 0.50 | 0.02 | $F_{(2,8385)} = 26.8$, ***p* < 0.01** | *ipcc-soto* & *soto-spawn* (***p* < 0.01**); *ipcc-spawn* (*p* = 0.98) |
| SOC | 0.23 | 0.02 | $F_{(1,5594)} = 0.3$, *p* = 0.556 | *esdac-olm* (*p* = 0.556) |

[a]**Bold** indicates statistically-significant values

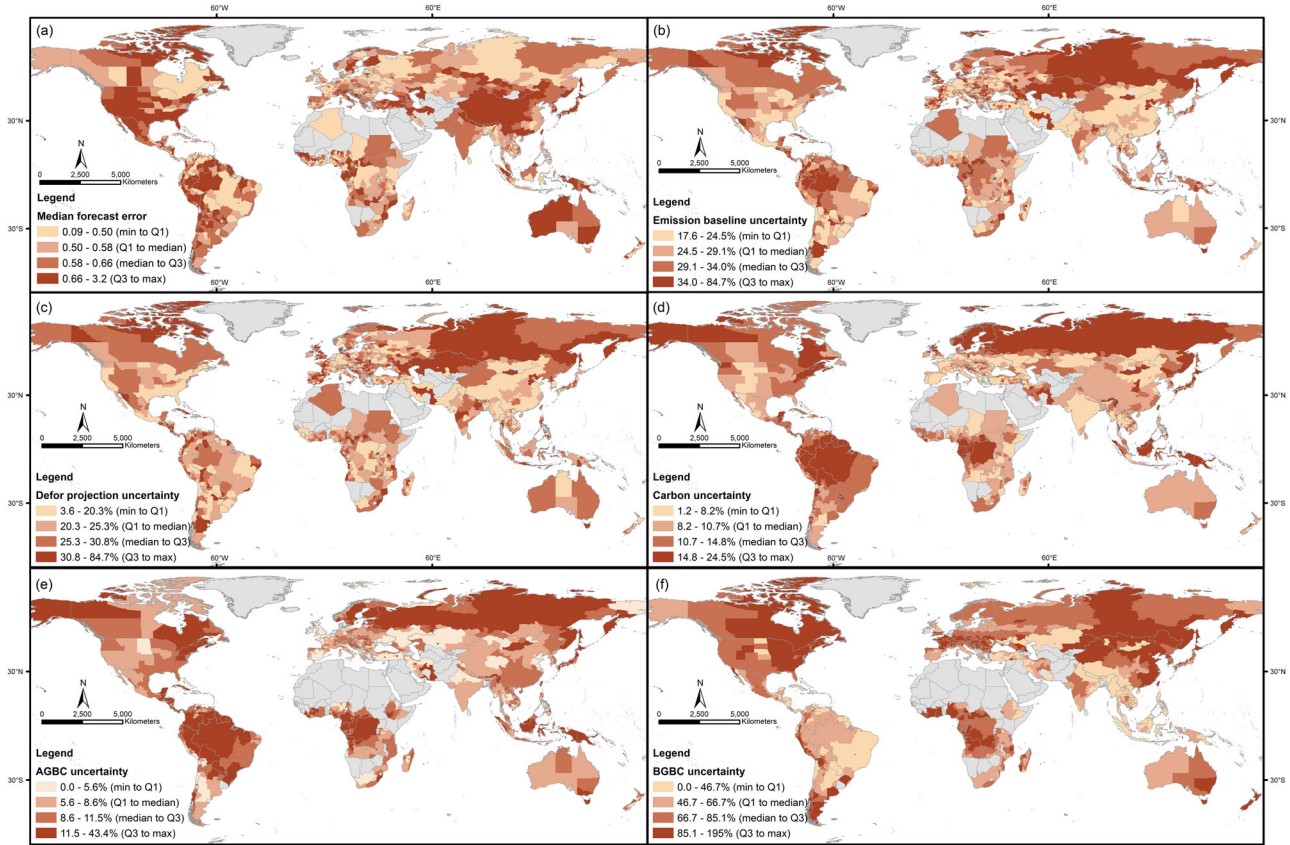

**Fig. 3 | Median forecast error and uncertainty in deforestation projections and deforestation emission baseline components across jurisdictions globally.** Global maps of national or subnational jurisdictions, with colour indicating median forecast error of different deforestation projection approaches **a**, and median relative uncertainty (90% CI) of deforestation emission baselines **b**, of deforestation projections **c**, of carbon estimates **d**, of AGBC estimates **e**, and of BGBC estimates **f**.

projects with more than 15% overall uncertainty to apply a discount, or may even prohibit their listing;[14] we found that a median of 28.0% (90% range: 20.0–60.0%) of carbon estimation methods exceeded 15% relative uncertainty for each jurisdiction, and a median of 52.9% (90% range: 48.9–58.5%) of deforestation projection methods exceeded 15% relative uncertainty for each jurisdiction, giving a propagated overall median of 68.0% (90% range: 61.1–85.6%) for all possible methods for each jurisdiction.

## Discussion

Deforestation emission baselines vary considerably depending on the methods and parameters used, within the range of commonly permissible methods and parameters. Many voluntary carbon standards also permit upward adjustments with justification, such as for High Forest Low Deforestation (HFLD) jurisdictions and in other circumstances, as flexibility for individual projects to make methodological choices[14,30,31]. Although such flexibility is intended to allow for

**Table 2 | Variable importance for how each parameter type influences the forecast error of predicted deforestation rates, as well as statistical tests (one-way ANOVA and Tukey's HSD post-hoc) comparing forecast errors between the different levels of each parameter, for n = 2794 jurisdictions**

| Parameter | Variable Importance | | Statistical tests | |
|---|---|---|---|---|
| | Mean | SE | One-Way ANOVA | Tukey's HSD post-hoc |
| Projection approach | 4.90 | 0.12 | $F_{(6,19424)} = 366$, $p < 0.01$ | All pairs $p < 0.01$, except between linear models global, global_s, regional, and regional_s ($p = 1.0$) |
| Forest dataset | 2.28 | 0.16 | $F_{(4,12646)} = 63$, $p < 0.01$ | All pairs $p < 0.05$, except between the hansen15-hansen30 ($p = 0.70$), hansen30-hansen60 ($p = 0.07$), and hansen60-esacci ($p = 0.76$) |
| Historical reference years | 0.82 | 0.42 | ANOVA $F_{(5,16600)} = 4.2$, $p < 0.01$ | 5–10, 6–10, 7–10, 8–10 ($p < 0.05$), all other pairs $p > 0.05$ |

<sup>a</sup>**Bold** indicates statistically-significant values

necessary adjustments to the actual unique local circumstances, it can make baseline setting a matter of political judgement, and opens up carbon market participants to potential credibility and reputational risks should these baselines later be independently assessed to be flawed[12,19]. Performing sensitivity analyses (such as those used in our study) to assess the variability of baselining methods and parameters permitted, can help guide efforts to overhaul carbon standards and increase transparency to market participants. Sensitivity analyses could also be required for carbon project or programme developers in order to justify methodological choices, and as an additional safeguard to ensure that the calculated baselines are conservative.

In spite of efforts to advance the methods and science behind deforestation projections, it remains challenging to accurately predict deforestation rates for the future, as deforestation rates are influenced by various stochastic political, socio-economic, and biophysical environmental factors[32,33]. We show that the choice of deforestation projection approach has the greatest influence on baselines, with linear regression models incorporating various driving factors performing better than simpler and more commonly-used historical average and time function models, which do not include such driving factors. Our results also showed that forest datasets had the second-largest influence on baselines. Given that these different approaches, models, and data sources have their own unique spatial variabilities and biases, multi-model and multi-source ensemble approaches to overcome these variabilities and biases can be considered best practice, similar to other modelling communities such as the climate modelling, disease modelling, and remote sensing communities[34–36]. However, the feasibility and costs required for enhanced modelling efforts, as well as the need for more elaborate reporting and decision-making processes when faced with contradicting baselines, may become barriers to implementation.

We found large uncertainties in most combinations of methods and parameters, exceeding the 15% typically allowed in many voluntary carbon market standards; however, actual project developers may have access to additional observational data which could help reduce these uncertainties. Prior research has observed that uncertainties in deforestation baselines are commonly underestimated[37]. With such large uncertainties in the baselining methods analysed by our study, cost-effective approaches to reduce uncertainties will be relevant to the carbon market, such as improving experimental design and effective use of statistical techniques in place of extensive fieldwork[38].

There are many potential areas for further fundamental scientific research to improve baselining methods. Given that forest dataset is the second most important variable affecting variabilities in baselines as well as historical accuracy, further work to improve forest datasets, such as by harmonising definitions of forests and ecosystems, and improving methods for quantifying their area and functional value, will be important[39–42]. With higher uncertainties in carbon estimation in the tropics, efforts to acquire more high-quality validation data and perform more forest research in the tropics, are also important. Further work on improving how permanence and leakage risks are monitored and managed may also need to consider the impact of inaccurate and uncertain baselines, as analysed in our study.

Carbon markets play a crucial role in financing nature-based climate solutions such as avoided deforestation and restoration projects, which can potentially provide up to one-third of cost-effective climate mitigation to meet the <2 °C target of the Paris Climate Agreement[8]. Our analyses show that baselining methods for avoided deforestation projects have high variabilities and uncertainties, and face challenges in accuracy; approaches such as sensitivity analyses and the use of multi-model and multi-source ensembles may help. These results can help guide carbon market participants in decision-making, and provide a useful roadmap for further research and practice to improve baseline estimation methods and thus enhance carbon market integrity and trust.

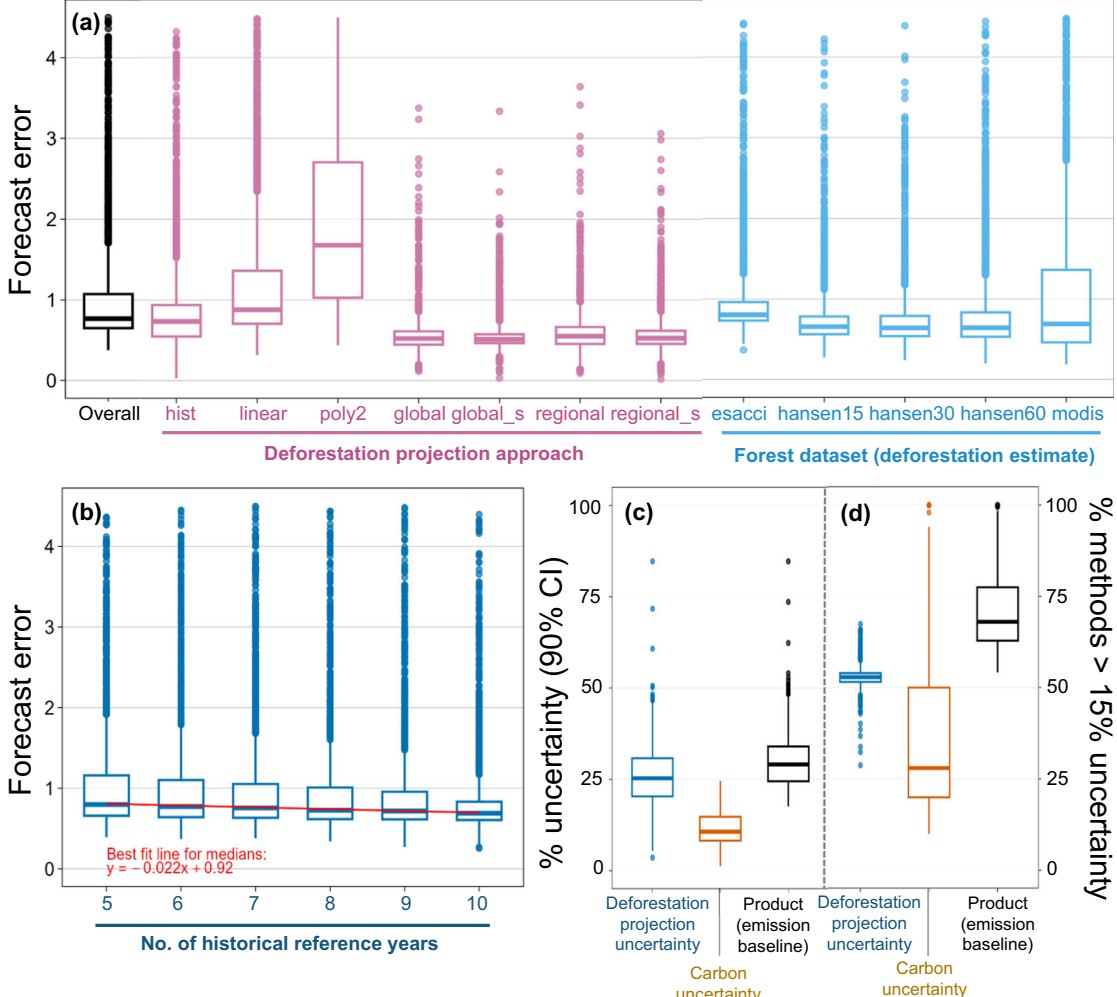

**Fig. 4 | Jurisdictional forecast error and uncertainty in deforestation emission baselines.** Box plots showing distribution of jurisdictional forecast error **a**, **b** of deforestation emission baselines (each point represents one jurisdiction, for $n = 2{,}794$ jurisdictions) for different component parameters (see explanation of these datasets and methods in Method section), as well as jurisdictional median relative uncertainty (90% CI) for carbon estimation, deforestation projection model, and propagated uncertainty for the deforestation emission baseline **c**, and the % of methods with > 15% uncertainty **d**.

## Methods

We first assessed the range of deforestation emission baselining methods commonly used for avoided deforestation carbon crediting by examining project documents for jurisdictional and site-based approaches. This included 30 VM0015 Verified Carbon Standard (VCS) projects, 4 Plan Vivo projects, and 55 Forest Reference Emissions Level (FREL) submissions to the United Nations Framework Convention on Climate Change (UNFCCC) by countries intending to implement activities under Reducing Emissions from Deforestation and Forest Degradation (REDD + ) (see Supplementary Data 1 and Note S1-2).

Based on this assessment, we used a high-performance computing cluster to calculate deforestation emission baselines for each jurisdiction using all unique combinations ( ~ $4 \times 10^9$) within the range of parameters and methods typically permitted and used in avoided deforestation carbon crediting (Table 3). Deforestation emission baselines are the product of the projected deforestation rate and average forest carbon in $CO_2$e per jurisdiction. These possible deforestation emission baselines were calculated for each of 2794 level-0 national jurisdictions or level-1 subnational jurisdictions[43]. Only nations with at least 1000 km$^2$ of forest (as estimated by MODIS for the year 2001) were included; nations with more than 10,000 km$^2$ of forest were divided into level 1 subnational jurisdictions, while subnational jurisdictions with less than 10% forest cover by land area were excluded.

The types of projection approaches used typically falls into three categories: (i) historical average, which is a continuation of the average annual rate calculated for the historical reference period; (ii) time function, where historical trends are extrapolated to the future from the historical reference period using a linear or logistic regression; and (iii) modelling, which uses a model that expresses future deforestation as a function of driver variables. In this study, we used the historical average, two types of time function (linear time function and 2$^{nd}$ order polynomial time function), as well as four sets of linear models. In the linear models, the response variable (deforestation rate) was box-cox transformed, with outliers removed. There were up to 12 explanatory driver variables to predict deforestation which were biophysical and socioeconomic covariates (Table 4, brief explanation and rationale in Note S3), derived from commonly-used spatial products available at global-scale. Note that in this study, remotely-sensed tree cover loss or forest loss are considered to be synonymous with deforestation; for consistency among all datasets, forest gain or forest regeneration are not included. These spatial datasets were processed in Google Earth Engine using Mollweide equal-area projection. Next, statistical modelling and calculations were performed in R version 4.0.2, using the statistical packages 'MASS' v7.3-51.6', 'dplyr' v1.1.2, and the parallelisation packages 'foreach' v1.5.0 and 'doParallel' v1.0.15. Bidirectional stepwise elimination was performed to determine variables that

**Table 3 | Parameters and range of methods, data sources, and values used in this study**

| Parameter | Range of methods, data sources, and values used in this study |
|---|---|
| Projection approach | Historical average |
| | Linear time function |
| | 2nd order polynomial time function |
| | Global linear model (all 12 driver variables; see Table 4) |
| | Global linear model (300 repetitions of 3–11 random combinations of driver variables) |
| | Regional linear model (all 12 driver variables; see Table 4) |
| | Regional linear model (300 repetitions of 3–11 random combinations of driver variables) |
| Forest dataset | European Space Agency Climate Change Initiative Land Cover (ESA CCI-LC) land cover 300 m, forest defined by land covers 50-90 and 160-170 where tree cover > 15%[25]. Forest cover for each year was clipped to the previous year to allow only forest loss. |
| | Hansen Global Forest Change 30 m, forest defined by tree cover > 15%[23] |
| | Hansen Global Forest Change 30 m, forest defined by tree cover > 30%[23] |
| | Hansen Global Forest Change 30 m, forest defined by tree cover > 60%[23] |
| | MODIS MCD12Q1.061 land cover 500 m, forest defined by land covers 1-6 under classification Type 1 (International Geosphere Biosphere Programme classes) where tree cover > 60%[24]. Forest cover for each year was clipped to the previous year to allow only forest loss. |
| Historical reference years | 5–15 years |
| Starting year | 2000–2021 (Hansen) |
| | 2001–2020 (MODIS) |
| | 1992–2020 (ESA-CCI) |
| Aboveground biomass (AGB) | Spawn et al.[27], 300 m |
| | Global Ecosystem Dynamics Investigation (GEDI) L4B[28], 25 m |
| | Soto-Navarro et al.[26], 300 m |
| Belowground biomass (BGB) | Spawn et al.[27], 300 m |
| | Soto-Navarro et al.[26], 300 m |
| | Intergovernmental Panel for Climate Change (IPCC) BGB ratios[45] |
| Soil organic carbon (SOC) | European Soil Data Centre (ESDAC)[46], 30 arc sec |
| | OpenLandMap[47], 250 m |

**Table 4 | Data sources and processing methods for biophysical, socioeconomic, and land cover explanatory variables**

| Variable | Data source and processing methods |
|---|---|
| Elevation | Gap-filled digital elevation data from Shuttle Radar Topography Mission (SRTM)[48] |
| Slope | |
| Long-term mean annual temperature | WorldClim BIO Variables V1[49] |
| Long-term mean annual precipitation | |
| Global gridded per capita Gross Domestic Product (GDP) adjusted for purchasing power parity (PPP) | Extracted for 1992–2015; for years beyond 2015, data from 2015 was used due to the unavailability of more recent data[50] |
| Human Development Index | |
| Average nightlight intensity | Average nightlight intensity extracted from consistent and corrected DMSP-OLS data for 1992-2013; VIIRS nightlights data was extracted for 2012–2020 and calibrated to DMSP-OLS as per the method of Li et al[51]. and Teo et al[52]. by spatially aggregating both VIIRS and DMSP-OLS to 1 km resolution, generating power regression models for the temporally overlapping mosaics, and using the average of the resulting coefficients to calibrate all VIIRS images[53,54]. |
| Population density | Extracted for 1992-2020 from the Global Human Settlement layer (GHSL)[55] |
| Percentage forest area | Derived from forest datasets as described in Table 3. |
| Percentage agricultural area | For consistency with forest datasets, we used: land cover classes 10 and 20 from the ESA CCI-LC where ESA CCI-LC was used for forest cover; Global Food Security-Support Analysis Data (GFSAD)[56] where Hansen Global Forest Change was used for forest cover; and MODIS MCD12Q1.061 land cover class 12 where MODIS was used for forest cover. |
| Percentage land area occupied by mining land uses | Extracted for the year 2019, and applied to all years, based on the assumption that areas mined in the year 2019 generally represent areas with easily accessible and known mineral deposits which present an incentive for deforestation, and also because mining infrastructure tends to remain for decades[57]. |
| Percentage land area occupied by tree plantations | Tree plantations are distinct from natural forest, and may consist of tree crops such as oil palm, coffee, or coconut, or planted forests grown for wood production. Extracted for 1992–2020 from global map of planting years of plantations[58]. |

created the best model fit. A set of global models were run: the first linear model used all 12 explanatory driver variables (before stepwise elimination), while the second linear model was the mean of 300 repetitions of 3-11 random combinations of these explanatory driver variables. Next, these linear models were repeated for six regions of the world separately: Africa, Asia, Oceania, Latin America, North America, and Europe. These seven projection approaches in our study were then applied for all possible combinations of the different forest datasets, number of historical reference years, and range of years available, to derive the projected deforestation rate. Only non-negative projected deforestation values were retained, as methods which generate negative projected deforestation values would likely be rejected in actual baselines.

Average forest carbon in $CO_2$e per jurisdiction was estimated from various aboveground biomass (AGB), belowground biomass (BGB), and soil organic carbon (SOC) datasets. These spatial datasets were processed in Google Earth Engine using Mollweide equal-area projection. All biomass was converted to carbon stock by a 0.47 stoichiometric factor, then to carbon dioxide equivalent emissions ($CO_2$e) by a 3.67 molar mass conversion factor. We assumed a conservative 10-year decay estimate for the belowground carbon pool[44].

Uncertainties were derived from the time function and linear models by extracting the 90% prediction intervals from the models, in line with ART TREES methodology[30] and IPCC convention[1]. Uncertainties for carbon pools were either derived from the data provider where available, or used the IPCC default of 33%. Finally, these uncertainties were propagated by summation of quadrature.

Finally, variable importance was assessed by using random forest machine learning models in the R package 'ranger' v0.13.1, with deforestation emission baselines as the response variable and type of parameter as the explanatory variables. To improve computational efficiency, we bootstrapped 100 random forest models with 200 trees each, and with each random forest model taking a random sample of 900,000 rows.

## Data availability
All data generated by this study is included in the manuscript and supplementary, as well as the Figshare repository (https://doi.org/10.6084/m9.figshare.24597315).

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

## Acknowledgements

L.P.K. is supported by the Singapore National Research Foundation (NRF-RSS2019-007), as well as the Singapore National Research Foundation and Agency for Science, Technology and Research LCER Phase 2 funding programme (U2303D4101). H.T. is supported by the Singapore Ministry of Education (MOE-T2EP50122-0006). T.V.S. is supported by MAC3 Impact Philanthropies. Q.Z. is supported by the Strategic Hiring Scheme Fund of the Hong Kong Polytechnic University (P0044791).

## Author contributions

H.C.T., L.P.K., D.A.F., H.T. conceived the study. L.P.K., D.A.F., H.T. supervised the study. H.C.T. conducted the analyses and led the writing. N.H.L.T. performed literature review and data collection. All authors (H.C.T., N.H.L.T., Q.Z., A.J.Y.L., R.S., X.C., Y.Z., T.V.S, J.D.T.D.A, H.T., D.A.F., L.P.K.) contributed to designing the methodologies, writing, and editing.

## Competing interests

The authors declare no competing interests.
