## [Peer Review File · Nature Communications]

Uncertainties in deforestation emission baseline methodologies and implications for carbon marketsREVIEWER COMMENTS

Reviewer #1 (Remarks to the Author):

This is an interesting piece of work looking at an important topic in deforestation reduction projects. The authors have assessed the variability, accuracy, and uncertainty of baselining methods on a range of typically-used methods and parameters. Their findings provide some discussion on how future studies can be improved by using sensitivity analysis, multi-model, and multi-source ensemble approaches.

I have a few comments on the paper for the authors' consideration:

I appreciate the efforts authors have put into deforestation projection. But I have a problem with the analysis though:

- i) The forest transition theory (Mather, 1992) describes changes in the forest stock in a country in relation to its level of development. The authors analyze and predict deforestation in several different regions. How do the authors deal with deforestation in countries at different stages of development?
- ii) The authors have 11 explanatory driver variables in a linear model to predict deforestation. Among the land cover variables, the authors considered the mining land uses and tree planting variables but did not include the agricultural land variable. Many previous studies have revealed that the expansion of agricultural land is an important driver of deforestation. For instance, over the period 1980–2000, more than 80% of new croplands were created at the expense of previously forested lands (Gibbs et al., 2010). The authors need to consider the agricultural variable or explain why this variable is not included.

The authors concluded that multi-model and multi-source approaches contribute to reducing uncertainties in baseline estimations. However, I am confused with the analysis though:

- i) I understand that the multi-model refers to "300 linear models which used random combinations of between 3 and 10 driver variables (model_s)". But I do not read the definition of the multi-source approach in the text, does it refer to a combination of data from different sources?
- ii) Do you conclude that the multi-model approach reduces uncertainty in baseline estimations because it reduces outliers in the results? Also, how do the authors conclude that a multi-source approach reduces uncertainty?

The use of "accuracy" is unclear. In statistical analysis, accuracy is described as the closeness of the observed value of a test indicator or trait in an experiment or survey to its true value. However, the accuracy in this paper is the relative difference between actual and predicted deforestation rates.

Line 35-37: Add information on the traded volume of REDD+ carbon credits to better illustrate that it is the most traded in the voluntary market.

Line 41-44: To what "standards" are you referring here? Please also explain why they are conservative.

Line 50-53: Consider adding a few typical baselining methods for less familiar readers.

Line 63: To what "principle" are you referring?

An extra semicolon in line 159.

In Figure 3, the legend in the lower left corner is not legible. The clarity of this figure should be improved.

Reviewer #2 (Remarks to the Author):

Reviewer Recommendation and Comments for Manuscript NCOMMS-23-34224-T "Uncertainties in deforestation emission baseline methodologies and implications for carbon markets "

The article examines the extent to which deforestation emission baselines vary because of the different details of the allowable baseline determination methods, the factors responsible for this variation, and the spatial patterns that result. This is an important question - not only from a scientific

point of view, but also due to its practical relevance for baseline determination, and thus ultimately for the effectiveness of climate policy in the context of REDD.

The manuscript is well and mostly clearly written, with a few minor inaccuracies and/or ambiguities. I suggest accepting it after some (mostly minor) changes.

Specific comments

- Line 22, “the most vulnerable” sounds a bit propagandistic and might not be substantiable. “Forest residents” or the like might be more neutral
- 89, “negative but significant correlation between forest area and relative variability”: According to Fig.S3, this was $\log(\text{FA})$ rather than FA (this may confuse a reader – and, as Pearson’s r rather than Spearman’s rank correlation coefficient has been calculated, it even makes a difference). – And why the “but”?
- Since the number of observations is very different for the different statistical tests presented, it is suggested to add “ $n=...$ ” where it fits (e.g. headings of Tab.1, 2, Fig. S3, ...)
- 199, “...tended to reduce median relative variabilities very slightly, although this was not statistically significant”: This is not wrong, but misleading, as the effect size is quite substantial (i.e. the variability is reduced to less than three quarters with 11 more reference years). The lack of significance is very probably due to a low number of observations (presumably, only 11 median values have been compared here). If this is correct, the sentence could read “A longer historical reference period, which ranged between 5 and 15 years in this study, reduced relative median variabilities up to more than 25%; however, due to the small number of median values compared here, this is not statistically significant”. (Cf. also the ANOVA results)
- Fig. 2a: The regression formula $y=-5.12x+194$ does not fit well to the data shown in the graph. It seems as if $y= f(x-4)$ had been calculated here, rather than $y=f(x)$. If this is correct, the formula should read $Y=-5.12x+214.48$
- 153, the median jurisdiction’s forecast error, or the jurisdictions’ median forecast error? (According to the abstract, line 8, it is neither of the two, but the median jurisdiction’s baseline, rather than its forecast error).
- “predicted deforestation was different from actual deforestation by 0.828 times of the actual deforestation”: This sentence is not easy to understand. Isn’t this rather “on average, the [confidence intervals of the] predicted deforestation values amounted to only 82.8% of [those of] the actual deforestation”? (see also abstract, line 9-10, where this is again put a bit differently). – If this is the case, this observation would also be worthy of some more thoughts on the possible reasons in the discussion section (see below)
- 159, delete “;”
- 172 unclear; suggestion: “Fig.3 Global map of national or subnational jurisdictions, with colour indicating median forecast error of different deforestation projection approaches (a), and median relative uncertainty (90% CI) of deforestation emission baselines (b), of deforestation projections (c), of carbon estimates (d), of AGBC estimates (e), and of BGBC estimates (f)”.
- Fig.4b, like before, the “best fit line for medians” might likely be $y=-0.022x+0.968$ (n.b. in the supplementary materials the respective regression formulas appear to be correct)
- 216, read “analyses”
- 229, “should be best practice”: as a recommendation for scientific analyses this is reasonable. However, if this was meant as a recommendation for policy, further problems would have to be discussed (e.g. the increased demands on time and resources associated with such a proposal, which could be counterproductive for a policy that aims at avoiding deforestation; the possible decision-making problems in the presence of different/contradictory baselines, for which a solution approach is lacking here, etc.)
- 238, “cost-effective approaches to reduce uncertainties will be relevant to the carbon market”: this is fairly general.
- 255 Two points that could (but need not) be additionally problematized in the discussion: First, the problem of calculating forecast errors by using actual deforestation rates for comparison, which may already be influenced by anti-deforestation policies; second, a more general integration of the problem

of uncertain deforestation rates into the problem environment when determining baselines (e.g., the leakage problem - which can thwart the goal of avoiding deforestation, especially in the case of project-based baselines).

- 293, only non-negative values were retained: It is not quite understandable why several maps nevertheless show positive deforestation rates, even for those countries where forest area development is positive, e.g. Fig. S1 (and additionally, why and how deforestation emission baselines have been calculated even for non-REDD countries, e.g. Fig.1, 3a)
- Tab.4, "years after used data from 2015": check sentence
- Tab.4, on variable selection and explanation: For all variables listed here, it would be informative to the readers if the reasoning for their selection were shortly explained (i.e. some theoretical background), and for the proxy variables (e.g. nightlight intensity), what they are supposed to approximate. "Percentage forest area" and "percentage land area occupied by tree plantations" likely have quite some overlap; these terms should be more clearly defined (and possible problems due to multicollinearity be discussed – e.g. the possible consequences for the stepwise elimination of insignificant variables).
- 338 (Domke et al. 2019 = IPCC 2019): authors Garcia-Apaza, Grassi, Gschwantner disappeared in abbreviations

Uncertainties in deforestation emission baseline methodologies and implications for carbon markets

Response to reviewers

We thank the reviewers for their positive comments, and appreciate the time taken to provide such detailed and thoughtful feedback. We are glad that the reviewers appreciate the important implications from our results on deforestation baseline methodologies.

In this revision, we have made substantial changes to all sections of the manuscript to address the reviewers' comments. We also performed additional analyses to improve the manuscript as per the reviewers' helpful comments. We are grateful to the reviewers for providing constructive feedback to help us improve the manuscript.

Below we provide a point by point response to the comments.

Reviewer #1

This is an interesting piece of work looking at an important topic in deforestation reduction projects. The authors have assessed the variability, accuracy, and uncertainty of baselining methods on a range of typically-used methods and parameters. Their findings provide some discussion on how future studies can be improved by using sensitivity analysis, multi-model, and multi-source ensemble approaches.

Response: We thank the reviewer for their positive comments. We appreciate the reviewer's thorough and constructive feedback.

I have a few comments on the paper for the authors' consideration:

I appreciate the efforts authors have put into deforestation projection. But I have a problem with the analysis though:

i) The forest transition theory (Mather, 1992) describes changes in the forest stock in a country in relation to its level of development. The authors analyze and predict deforestation in several different regions. How do the authors deal with deforestation in countries at different stages of development?

ii) The authors have 11 explanatory driver variables in a linear model to predict deforestation. Among the land cover variables, the authors considered the mining land uses and tree planting variables but did not include the agricultural land variable. Many previous studies have revealed that the expansion of agricultural land is an important driver of deforestation. For instance, over the period 1980–2000, more than 80% of new croplands were created at the expense of previously forested lands (Gibbs et al., 2010). The authors need to consider the agricultural variable or explain why this variable is not included.

Response: We thank their reviewer for their comments, highlighting how forest transition theory describes the relation between the stages in development (which can be reflected by agricultural land) and forest cover. For our linear modelling of driver variables, we had included variables which are proxies for developmental stage, such as Global gridded per capita Gross Domestic Product (GDP) adjusted for purchasing power parity (PPP), the Human Development Index (HDI), average

nightlight intensity, and population density. We appreciate the reviewer for pointing out agricultural land as an important variable, which is both an important deforestation driver and underpins the forest transition theory. We have added agricultural land as one of the variables for projecting forest loss, and re-run the entire analyses with this additional variable (LL325-326):

Table 4. Data sources and processing methods for biophysical, socioeconomic, and land cover explanatory variables.

[...]

Percentage agricultural area	For consistency with forest datasets, we used: land cover classes 10 and 20 from the ESA CCI-LC where ESA CCI-LC was used for forest cover; Global Food Security-Support Analysis Data (GFSAD) (Oliphant et al., 2022) where Hansen Global Forest Change was used for forest cover; and MODIS MCD12Q1.061 land cover class 12 where MODIS was used for forest cover.
--

We carefully considered the issue of how to deal with different regions of the world. Although variables which are proxies for developmental stage are already included in the global model and thus have already captured some inter-regional differences, we decided to add regional models (Africa, Asia, Oceania, Latin America, North America, and Europe) in addition to the global model of driver variables, following a similar approach used by Hewson et al. (2019) (LL309-311):

“Next, these linear models were repeated for six regions of the world separately: Africa, Asia, Oceania, Latin America, North America, and Europe.”

Interestingly, the regional models were more variable than the global models, and were no more accurate than the global models.

(LL1211-123) “However, the regional models had significantly higher median relative variabilities (*regional*: 70.7%, *regional_s*: 65.4%) than their corresponding global models (*global*: 66.0%, *global_s*: 61.0%).”

(LL166-168) “Forecast errors for the linear models (*global*, *global_s*, *regional*, and *regional_s*) were significantly lower than other approaches, but there were no significant differences between the different linear models.”

Hewson, J., Crema, S. C., González-Roglich, M., Tabor, K., & Harvey, C. A. (2019). New 1 km Resolution Datasets of Global and Regional Risks of Tree Cover Loss. *Land*, 8(1), 14. <https://doi.org/10.3390/LAND8010014>

The authors concluded that multi-model and multi-source approaches contribute to reducing uncertainties in baseline estimations. However, I am confused with the analysis though:

i) I understand that the multi-model refers to “300 linear models which used random combinations of between 3 and 10 driver variables (*model_s*)”. But I do not read the definition of the multi-source approach in the text, does it refer to a combination of data from different sources?

Response: We thank the reviewer for their clarification. Here, we use the term “multi-model” to refer to the “300 linear models of random combinations” as well as the other different categories of models (namely historical average and time function models). We use “multi-source” to refer to data sources, in particular forest datasets, as we have examined the effect of using different forest

datasets at different resolutions in this study. We have rewritten the relevant sentences to be clearer on our intention (LL236-244):

“We show that the choice of deforestation projection approach has the greatest influence on baselines, with linear regression models incorporating various driving factors performing better than simpler and more commonly-used historical average and time function models, which do not include such driving factors. Our results also showed that forest datasets had the second-largest influence on baselines. Given that these different approaches, models, and data sources have their own unique spatial variabilities and biases, multi-model and multi-source ensemble approaches to overcome these variabilities and biases should be the best practice, similar to other modelling communities such as the climate modelling, disease modelling, and remote sensing communities (Reich et al., 2019; Tebaldi & Knutti, 2007; Zhang, 2010).”

ii) Do you conclude that the multi-model approach reduces uncertainty in baseline estimations because it reduces outliers in the results? Also, how do the authors conclude that a multi-source approach reduces uncertainty?

Response: Yes, our results suggest that a multi-model approach can reduce the relative variabilities of different baselining methods in each jurisdiction, as the multi-model ensemble (300 linear models) “reduces the interquartile range (IQR) as well as outliers among jurisdictions” (LL120-121). Moreover, with forecast errors (LL164-170):

“Each projection approach had different spatial variabilities and biases in how accurately they predicted deforestation rates (Fig. S4), without there being clear regional patterns. Forecast errors for the linear models (*global*, *global_s*, *regional*, and *regional_s*) were significantly lower than other approaches, but there were no significant differences between the different linear models. However, by taking the median of all approaches, it is possible to overcome the individual spatial variabilities and biases that are inherent to each individual projection approach.”

As for a multi-source approach, different spatial variabilities and biases can also be observed (Figure S4). Although in this study we did not directly test a multi-model ensemble of different model types (historical average, time function, and linear modelling) nor a multi-source ensemble, following the same principles we may expect to also reduce variabilities and biases from ensemble approaches. This can be further investigated by future studies.

The use of “accuracy” is unclear. In statistical analysis, accuracy is described as the closeness of the observed value of a test indicator or trait in an experiment or survey to its true value. However, the accuracy in this paper is the relative difference between actual and predicted deforestation rates.

Response: We thank the reviewer for their comment on the definition of “accuracy” in statistical analysis. Here we use the term “accuracy” in a similar way to the remote sensing discipline, where “accuracy assessments” are conducted by comparing predicted values against actual data which are assumed to approximate the true values (Foody, 2002).

Foody, G. (2002). Status of land cover classification accuracy assessment. *Remote Sensing of Environment*, 80(1). [https://doi.org/10.1016/S0034-4257\(01\)00295-4](https://doi.org/10.1016/S0034-4257(01)00295-4)

Line 35-37: Add information on the traded volume of REDD+ carbon credits to better illustrate that it is the most traded in the voluntary market.

Response: We thank the reviewer for their suggestion. We have added this information (LL35-37):

“In recent years, voluntary carbon markets have expanded rapidly to reach over \$2 billion by 2021, with REDD+ projects being the largest by traded volume at \$863 million (Donofrio et al., 2022).”

Line 41-44: To what “standards” are you referring here? Please also explain why they are conservative.

Response: We thank the reviewer for their clarification. We have rephrased the relevant sentences to explain which standards are referred to here, and why they are conservative (LL37-52):

“However, concerns regarding the credibility of avoided deforestation carbon credits have once again risen to the forefront (Balmford et al., 2023), as contrasting evidence on their effectiveness has cast doubt on the methods used to estimate deforestation emission baselines, which define the business-as-usual scenario upon which the emission reductions and subsequent carbon credits are calculated (Guizar-Coutiño et al., 2022; West et al., 2020). This has posed challenges for carbon market standards, which include the Verified Carbon Standard (VCS), Forest Carbon Partnership Facility (FCPF), and the Architecture for Reducing Emissions from Deforestation and Forest Degradation (REDD+) Transactions: The REDD+ Environmental Excellence Standard (ART TREES), among others. To maintain credibility, carbon market standards are typically designed to be conservative, through mechanisms such as selecting methodologies with conservative assumptions, and requiring deductions for permanence, leakage, and uncertainty (Chagas et al., 2020; Grassi et al., 2008). Although overestimated baselines may generate credits that lack actual emission reductions, thereby resulting in inefficient resource allocation, underestimated baselines may result in insufficient financial incentives for forest protection. Thus, uncertainties in baseline projections have significant implications for avoided deforestation schemes (Friess & Webb, 2014).”

Line 50-53: Consider adding a few typical baselining methods for less familiar readers.

Response: We thank the reviewer for their suggestion. We have added examples of carbon standards which use these baselining methods (LL53-55):

“Avoided deforestation projects are required to calculate deforestation emission baselines and their corresponding uncertainties following prescribed methods and guidelines (see Methods for details), such as VCS, FCPF and ART TREES standards in the voluntary carbon market.”

A more detailed description of baselining methods is available in the Methods section (LL291-295):

“The types of projection approaches used typically falls into three categories: (i) historical average, which is a continuation of the average annual rate calculated for the historical reference period; (ii) time function, where historical trends are extrapolated to the future from the historical reference period using a linear or logistic regression; and (iii) modelling, which uses a model that expresses future deforestation as a function of driver variables.”

Line 63: To what “principle” are you referring?

Response: We have further amended this to “broad methodological principles” (LL70). This would include the fundamental approach to baselines as mentioned earlier in the text, such as that “Typically, the deforestation emission baseline is a product of two components – a projected deforestation rate for the future period and a forest carbon estimate” (LL55-57), as well as “To

maintain credibility, carbon market standards are typically designed to be conservative, through mechanisms such as selecting methodologies with conservative assumptions, and requiring deductions for permanence, leakage, and uncertainty” (LL45-48).

An extra semicolon in line 159.

Response: We thank the reviewer. We have removed the extra semicolon.

In Figure 3, the legend in the lower left corner is not legible. The clarity of this figure should be improved.

Response: We thank the reviewer. We have increased the font size and improved the figure’s resolution.

Reviewer #2

The article examines the extent to which deforestation emission baselines vary because of the different details of the allowable baseline determination methods, the factors responsible for this variation, and the spatial patterns that result. This is an important question - not only from a scientific point of view, but also due to its practical relevance for baseline determination, and thus ultimately for the effectiveness of climate policy in the context of REDD.

The manuscript is well and mostly clearly written, with a few minor inaccuracies and/or ambiguities. I suggest accepting it after some (mostly minor) changes.

Response: We thank the reviewer for their positive comments. We appreciate the reviewer’s thorough and constructive feedback.

Specific comments

- Line 22, “the most vulnerable” sounds a bit propagandistic and might not be substantiable. “Forest residents” or the like might be more neutral

Response: We thank the reviewer for their comment. We have changed to the term “forest-dwelling communities” (LL22-23).

- 89, “negative but significant correlation between forest area and relative variability”: According to Fig.S3, this was $\log(\text{FA})$ rather than FA (this may confuse a reader – and, as Pearson’s r rather than Spearman’s rank correlation coefficient has been calculated, it even makes a difference). – And why the “but”?

Response: We thank the reviewer for their observation. We have revised the sentence for clarity (LL96-97):

“There was a significant negative correlation between log-transformed forest area and relative variability, with Pearson’s $r = -0.34$ and $p < 0.01$ (Fig. S3a)”

- Since the number of observations is very different for the different statistical tests presented, it is suggested to add “n=...” where it fits (e.g. headings of Tab.1, 2, Fig. S3, ...)

Response: We thank the reviewer for their suggestion. We have added sample sizes (n) for clarity:

(LL142-144) “Table 1. Variable importance for how each parameter type influences the deforestation emission baseline, as well as statistical tests (one-way ANOVA and Tukey’s

HSD post-hoc) comparing relative variabilities between the different levels of each parameter, for n = 2,794 jurisdictions.”

(LL177-179) “Table 2. Variable importance for how each parameter type influences the forecast error of predicted deforestation rates, as well as statistical tests (one-way ANOVA and Tukey’s HSD post-hoc) comparing forecast errors between the different levels of each parameter, for n = 2,794 jurisdictions.”

“Fig S3. (a) Forest area (log-10 adjusted) against % variability (CV) for each of n = 2,794 jurisdictions.”

“Fig S3. (b) Forest area (log-10 adjusted) against forecast error for each of n = 2,794 jurisdictions.”

“Fig S3. (c) Forest area (log-10 adjusted) against % uncertainty (90% CI) for each of n = 2,794 jurisdictions.”

- 199, “...tended to reduce median relative variabilities very slightly, although this was not statistically significant”: This is not wrong, but misleading, as the effect size is quite substantial (i.e. the variability is reduced to less than three quarters with 11 more reference years). The lack of significance is very probably due to a low number of observations (presumably, only 11 median values have been compared here). If this is correct, the sentence could read “A longer historical reference period, which ranged between 5 and 15 years in this study, reduced relative median variabilities up to more than 25%; however, due to the small number of median values compared here, this is not statistically significant”. (Cf. also the ANOVA results)

Response: We thank the reviewer for their suggestions. With the new analysis (adding agricultural land as a variable and including additional regional models), there have been slight changes to the results and these results are now significant. We agree with the reviewer that the insignificance of the statistical test in the earlier analysis was probably due to low sample size. The text now reads (LL128-131):

“A longer historical reference period, which ranged between 5 and 15 years in this study, also significantly reduced median relative variabilities (Table 1, Fig. 2b); this suggests that longer historical reference period can reduce the effect of stochastic variations in deforestation rates.”

- Fig. 2a: The regression formula $y = -5.12x + 194$ does not fit well to the data shown in the graph. It seems as if $y = f(x-4)$ had been calculated here, rather than $y = f(x)$. If this is correct, the formula should read $Y = -5.12x + 214.48$

Response: The reviewer is right that the regression formula was mistakenly calculated as $y = f(x-4)$. We have corrected our code and re-generated the plot and regression formula. We thank the reviewer for pointing out.

- 153, the median jurisdiction’s forecast error, or the jurisdictions’ median forecast error? (According to the abstract, line 8, it is neither of the two, but the median jurisdiction’s baseline, rather than its forecast error).

Response: We thank the reviewer for clarifying. For each jurisdiction, we first assessed the forecast error for all methods, then took the median of those to derive the median forecast error for that jurisdiction. Next, we assessed the median forecast errors for all jurisdictions, then took the median

of those to derive the median jurisdiction's median forecast error. We have rephrased for greater clarity:

(LL161-162) "The median jurisdiction's median forecast error between actual and predicted deforestation was 0.778 (90% range: 0.548-3.56) across all methods"

In the abstract, the sentence refers to the "median jurisdiction", describing its "deforestation emission baseline" in the first part, then the "median forecast error" in the second part.

(LL8-11): "The median jurisdiction's deforestation emission baseline varied by 171% (90% range: 87%-440%) of its mean, with a median forecast error of 0.778 times (90% range: 0.548-3.56) the actual deforestation rate."

- "predicted deforestation was different from actual deforestation by 0.828 times of the actual deforestation": This sentence is not easy to understand. Isn't this rather "on average, the [confidence intervals of the] predicted deforestation values amounted to only 82.8% of [those of] the actual deforestation"? (see also abstract, line 9-10, where this is again put a bit differently). – If this is the case, this observation would also be worthy of some more thoughts on the possible reasons in the discussion section (see below)

Response: We thank the reviewer for their comments. We were attempting to explain the concept of "forecast error" in a simpler way; we calculated "forecast error" using $\frac{|predicted-actual|}{actual}$. There are no confidence intervals involved; the 90% range refers to the range of possible low to high values from 90% of all jurisdictions (0.548-3.56) to compare against the median jurisdiction (0.778). Although we think that reporting forecast error using either % or a proportion (decimal) value would both be acceptable, we chose to use the proportion (decimal) value to avoid potential misunderstanding that accuracy/forecast error should be a maximum of 100% or 1, which is not the case here based on our definition of "forecast error". We have rewritten the sentence now as follows (LL161-164):

"The median jurisdiction's median forecast error between actual and predicted deforestation (calculated by $\frac{|predicted-actual|}{actual}$) was 0.778 (90% range: 0.548-3.56) across all methods (Fig. 3a), i.e. median predicted deforestation was different from actual deforestation by 0.778 times of the actual deforestation."

We hope that this presentation is clearer and welcome any additional suggestions from the reviewers or editors.

- 159, delete ";

Response: We thank the reviewer. We have removed the extra semicolon.

- 172 unclear; suggestion: "Fig.3 Global map of national or subnational jurisdictions, with colour indicating median forecast error of different deforestation projection approaches (a), and median relative uncertainty (90% CI) of deforestation emission baselines (b), of deforestation projections (c), of carbon estimates (d), of AGBC estimates (e), and of BGBC estimates (f)".

Response: We thank the reviewer for their suggestion. We like it and have rephrased as suggested (LL183-186).

- Fig.4b, like before, the "best fit line for medians" might likely be $y=-0.022x+0.968$ (n.b. in the supplementary materials the respective regression formulas appear to be correct)

Response: The reviewer is right that the regression formula was mistakenly calculated as $y = f(x-4)$. We have corrected our code and re-generated the plot and regression formula. We thank the reviewer for pointing out.

- 216, read “analyses”

Response: We thank the reviewer for pointing out and have revised as suggested (LL227).

- 229, “should be best practice”: as a recommendation for scientific analyses this is reasonable. However, if this was meant as a recommendation for policy, further problems would have to be discussed (e.g. the increased demands on time and resources associated with such a proposal, which could be counterproductive for a policy that aims at avoiding deforestation; the possible decision-making problems in the presence of different/contradictory baselines, for which a solution approach is lacking here, etc.)

Response: We thank the reviewer for highlighting this. Certainly, this is primarily a recommendation for scientific analysis and what the ‘ideal’ may be for actual carbon standards. We will bear this in mind when making policy recommendations in real-life. We have added a sentence to summarise these points (LL244-246):

“However, the feasibility and costs required for enhanced modelling efforts, as well as the need for more elaborate reporting and decision-making processes when faced with contradicting baselines, may become barriers to implementation.”

- 238, “cost-effective approaches to reduce uncertainties will be relevant to the carbon market”: this is fairly general.

Response: We thank the reviewer for their comment. We have revised and elaborated upon the sentence (LL251-254):

“With such large uncertainties in the baselining methods analysed by our study, cost-effective approaches to reduce uncertainties will be relevant to the carbon market, such as improving experimental design and effective use of statistical techniques in place of extensive fieldwork (Emick et al., 2023).”

- 255 Two points that could (but need not) be additionally problematized in the discussion: First, the problem of calculating forecast errors by using actual deforestation rates for comparison, which may already be influenced by anti-deforestation policies; second, a more general integration of the problem of uncertain deforestation rates into the problem environment when determining baselines (e.g., the leakage problem - which can thwart the goal of avoiding deforestation, especially in the case of project-based baselines).

Response: We thank the reviewer for raising these two very insightful points. For the first point, we agree with the reviewer that anti-deforestation policies are already influencing the actual deforestation rate. Since existing anti-deforestation policies (or even the same project/programme, depending on context) will have already influenced the existing deforestation rate, regardless of the method for predicting future deforestation (historical averages, time function, modelling) – as long as some reference to the existing deforestation rate is being made, this can introduce issues relating to additionality. These additionality issues would apply to both forecast errors (calculated in our study) and the baselines themselves. Given the complexity of additionality issues, we think it would be more suitable for a more thorough treatment in other studies.

For the second point, we have added this to the manuscript (LL320-321):

“Further work on improving how permanence and leakage risks are monitored and managed may also need to consider the impact of inaccurate and uncertain baselines, as analysed in our study.”

We appreciate the reviewer’s insights and hope that further research can more thoroughly problematize and hopefully find better solutions to manage the issues of additionality, permanence, and leakage in carbon markets.

- 293, only non-negative values were retained: It is not quite understandable why several maps nevertheless show positive deforestation rates, even for those countries where forest area development is positive, e.g. Fig. S1 (and additionally, why and how deforestation emission baselines have been calculated even for non-REDD countries, e.g. Fig.1, 3a)

Response: We thank the reviewer for their comment. We further clarify the reason why only non-negative projected deforestation values were retained (LL313-315):

“Only non-negative projected deforestation values were retained, as methods which generate negative projected deforestation values would likely be rejected in actual baselines.”

Our map showing actual deforestation rates (Fig. S1) only has non-negative (positive) values. The reviewer is right in pointing out that some jurisdictions have had a net forest gain, and if forest gain were included in the calculation, these jurisdictions would have negative deforestation rates. In our study, we do not include pixels which showed a forest gain, to ensure consistency with the Hansen Global Forest Watch dataset which only shows forest loss values from 2001-2022 with forest gain only from 2001-2012 and discontinued thereafter. Hence, our deforestation or forest loss figure, is not a net forest loss. We have added this explanation in the Methods (LL300-303):

“Note that in this study, remotely-sensed tree cover loss or forest loss are considered to be synonymous with deforestation; for consistency among all datasets, forest gain or forest regeneration are not included.”

This is also explained in Table 3, for the ESA-CCI LC and MODIS datasets (LL323):

“Forest cover for each year was clipped to the previous year to allow only forest loss.”

- Tab.4, “years after used data from 2015”: check sentence

Response: We thank the reviewer for their comment. We have revised the sentence as follows (LL325-326):

“Extracted for 1992-2015; for years beyond 2015, data from 2015 was used due to the unavailability of more recent data”

- Tab.4, on variable selection and explanation: For all variables listed here, it would be informative to the readers if the reasoning for their selection were shortly explained (i.e. some theoretical background), and for the proxy variables (e.g. nightlight intensity), what they are supposed to approximate. “Percentage forest area” and “percentage land area occupied by tree plantations” likely have quite some overlap; these terms should be more clearly defined (and possible problems due to multicollinearity be discussed – e.g. the possible consequences for the stepwise elimination of insignificant variables).

Response: We thank the reviewer for their helpful suggestions. We have added explanations and rationale for these variables as Supplementary Note S3. We have also added a definition for tree plantations (from the dataset source) in Table 4 (LL326):

“Tree plantations are distinct from natural forest, and may consist of tree crops such as oil palm, coffee, or coconut, or planted forests grown for wood production.”

We agree with the reviewer on the possibility of multicollinearity between some of these variables. We have looked through the literature and have noted many similar variables used by studies of deforestation drivers, as well as similar linear regression approaches, such as Hewson et al. (2019) and others. As the focus of this study is on comparing different baselining methods and parameters typically allowed, we focused on creating a broad range of possible methods and parameters, such as bootstrapping 300 repetitions of linear regression models, rather than refining the models to find the best model. Also, linear regression models are only one of the various types of models we used (e.g. historical average, linear time function and 2nd order polynomial time function).

Aragão, L. E. O. C., Malhi, Y., Barbier, N., Lima, A., Shimabukuro, Y., Anderson, L., & Saatchi, S. (2008). Interactions between rainfall, deforestation and fires during recent years in the Brazilian Amazonia. *Philosophical Transactions of the Royal Society B: Biological Sciences*, 363(1498), 1779–1785. <https://doi.org/10.1098/RSTB.2007.0026>

Bax, V., & Francesconi, W. (2018). Environmental predictors of forest change: An analysis of natural predisposition to deforestation in the tropical Andes region, Peru. *Applied Geography*, 91, 99–110. <https://doi.org/10.1016/j.apgeog.2018.01.002>

Bhattarai, K., Conway, D., & Yousef, M. (2009). Determinants of deforestation in Nepal's Central Development Region. *Journal of Environmental Management*, 91(2), 471–488. <https://doi.org/10.1016/j.jenvman.2009.09.016>

Carvalho Lima, T., Carvalho Ribeiro, S., & Soares-Filho, B. (2018). Integrating Econometric and Spatially Explicit Dynamic Models to Simulate Land Use Transitions in the Cerrado Biome. 399–417. https://doi.org/10.1007/978-3-319-60801-3_19

Grau, R. H., Gasparri, N. I., & Aide, M. T. (2005). Agriculture expansion and deforestation in seasonally dry forests of north-west Argentina. *Environmental Conservation*, 32(2), 140–148.

Hewson, J., Crema, S. C., González-Roglich, M., Tabor, K., & Harvey, C. A. (2019). New 1 km Resolution Datasets of Global and Regional Risks of Tree Cover Loss. *Land*, 8(1), 14. <https://doi.org/10.3390/LAND8010014>

Laurance, W. F., Albernaz, A. K. M., Schroth, G., Fearnside, P. M., Bergen, S., Venticinqu, E. M., & Da Costa, C. (2002). Predictors of Deforestation in the Brazilian Amazon. *Journal of Biogeography*, 28(5/6), 737–748. <https://www.jstor.org/stable/827480>

• 338 (Domke et al. 2019 = IPCC 2019): authors Garcia-Apaza, Grassi, Gschwantner disappeared in abbreviations

Response: We thank the reviewer for pointing out. We have corrected the reference (LL359-365):

Domke, G., Brandon, A., Diaz-Lasco, R., Federici, S., Garcia-Apaza, E., Grassi, G., Gschwantner, T., Herold, M., Hirata, Y., Kasimir, Å., Kinyanjui, M. J., Krisnawati, H., Lehtonen, A., Malimbwi, R. E., Niinistö, S., Ogle, S. M., Paul, T., Ravindranath, N. H., Rock, J., ... Zhu, J. (2019). Chapter 4: Forest Land. In *2019 Refinement to the 2006 IPCC Guidelines for National Greenhouse Gas Inventories Volume 4: Agriculture, Forestry and Other Land Use*. https://www.ipcc-nggip.iges.or.jp/public/2019rf/pdf/4_Volume4/19R_V4_Ch04_Forest_Land.pdf

REVIEWERS' COMMENTS

Reviewer #1 (Remarks to the Author):

Thank the authors for the changes they made to the article. I recommend this manuscript can be accepted.

Reviewer #2 (Remarks to the Author):

Reviewer Recommendation and Comments for revised manuscript NCOMMS-23-34224A "Uncertainties in deforestation emission baseline methodologies and implications for carbon markets"

The manuscript analyses an important and interesting topic and is well and clearly written; in their revision the authors have (in my opinion) also very convincingly cleared up the minor ambiguities that had previously existed. My suggestion is to accept the article.

I congratulate the authors on their good work and would like to express my appreciation for how thoroughly they revised the article, and also explained these revisions to the reviewers!

(Just as an aside: perhaps I owe the authors an explanation as to why I was a little confused about the definition of "tree plantations" in the penultimate comment of my last review: while a land use change from [primary] forest to "tree plantations" consisting of oil and coconut palms undoubtedly constitutes deforestation - especially since botanically speaking palms are not trees at all - there may be more significant demarcation problems with managed forests, which may also affect the baseline data used for this study. However, this demarcation problem plays only a comparatively minor role in the countries that are the focus of the REDD discussion, so that it is also of minor importance for this article).

Uncertainties in deforestation emission baseline methodologies and implications for carbon markets

Response to reviewers

We thank the reviewers for their positive comments, and appreciate the time taken to provide such detailed and thoughtful feedback which have helped to improve the manuscript.

Below we provide a point by point response to the comments.

Reviewer #1

Thank the authors for the changes they made to the article. I recommend this manuscript can be accepted.

Response: We thank the reviewer for their thorough and constructive feedback.

Reviewer #2

The manuscript analyses an important and interesting topic and is well and clearly written; in their revision the authors have (in my opinion) also very convincingly cleared up the minor ambiguities that had previously existed. My suggestion is to accept the article.

I congratulate the authors on their good work and would like to express my appreciation for how thoroughly they revised the article, and also explained these revisions to the reviewers!

(Just as an aside: perhaps I owe the authors an explanation as to why I was a little confused about the definition of "tree plantations" in the penultimate comment of my last review: while a land use change from [primary] forest to "tree plantations" consisting of oil and coconut palms undoubtedly constitutes deforestation - especially since botanically speaking palms are not trees at all - there may be more significant demarcation problems with managed forests, which may also affect the baseline data used for this study. However, this demarcation problem plays only a comparatively minor role in the countries that are the focus of the REDD discussion, so that it is also of minor importance for this article).

Response: We thank the reviewer for their thorough and constructive feedback. We also appreciate the additional explanation, which indeed add valuable context to our study.